# Prescription of Levofloxacin and Moxifloxacin in Select Hospitals in Uganda: A Pilot Study to Assess Guideline Concordance

**DOI:** 10.3390/antibiotics9080439

**Published:** 2020-07-23

**Authors:** Victoria Nambasa, Helen B. Ndagije, Allan Serwanga, Leonard Manirakiza, Joanitah Atuhaire, Diana Nakitto, Ronald Kiguba, Albert Figueras

**Affiliations:** 1Directorate of Product Safety, National Drug Authority (NDA), Kampala 23096, Uganda; hndagije@nda.or.ug (H.B.N.); aserwanga@nda.or.ug (A.S.); leonardmanirakiza16@gmail.com (L.M.); jatuhaire@nda.or.ug (J.A.); dnakitto@nda.or.ug (D.N.); 2Department of Pharmacology and Therapeutics, Makerere University College of Health Sciences, Kampala 7072, Uganda; kiguba@gmail.com; 3Departament de Farmacologia, Terapèutica i Toxicologia, Universitat Autònoma de Barcelona, 08193 Bellaterra, Spain; albert.figueras@gmail.com

**Keywords:** fluoroquinolones, multidrug-resistant tuberculosis, drug utilization study, prescription habits

## Abstract

*Background*: In Uganda, national tuberculosis (TB) treatment guidelines were revised to include the newer generation fluoroquinolones among the second-line treatment options for multidrug-resistant TB. This study was designed to analyze if the prescription of these quinolones is compliant with country recommendations. *Methods*: This was an observational retrospective study of consumption data for 2017 and 2018 across four selected regional referral hospitals. The sources of consumption data were hospital pharmacy stock cards and the dispensing register. The medical files of patients who had been prescribed fluoroquinolones were also assessed to study compliance with the Uganda Clinical Guidelines and the British National Formulary (BNF). *Results*: None of the 371 levofloxacin prescriptions analyzed complied with the Uganda Clinical Guidelines, although 250 (67.3%) were prescribed for indications included in the BNF. According to WHO prescription indicators, only 220 (59.3%) prescriptions were appropriate. *Conclusion*: The prescription of levofloxacin and moxifloxacin increased in the hospitals studied, but in a high proportion of cases, they were not compliant with country recommendations. The findings call for the strengthening of national antimicrobial stewardship programs.

## 1. Introduction

Tuberculosis (TB) is one of the world’s leading causes of adult morbidity and mortality [1]. Despite a steady decline in TB-related morbidity and mortality in Uganda, reports from the Ministry of Health (MoH) and the National TB and Leprosy Program (NTLP) indicate a high national TB burden at approximately 401 cases per 100,000 inhabitants [2]. There is also an emerging public health epidemic of multi-drug resistant and extensively drug-resistant TB (MDR/XDR-TB), which requires more focus on TB treatment. Thus, Uganda’s Clinical Guidelines (UCG) were revised to incorporate fluoroquinolones, particularly the newer-generation fluoroquinolones (levofloxacin, moxifloxacin, and gatifloxacin), as core medicines in both first and second-line treatment for MDR/XDR-TB; the use of these antibiotics for other clinical indications is discouraged [3,4].

The widespread use of antibiotics contributes to the emergence of antimicrobial resistance [5,6,7], with far-reaching consequences for patients with MDR/XDR-TB if fluoroquinolone resistance occurs [8], and infections associated with resistant microorganisms result in increased morbidity and mortality [9]. A recent study analyzing the pattern of point mutations in 80 isolates of *M. tuberculosis* from Eswatini, Somalia, and Uganda found that the frequency of fluoroquinolone resistance mutations is high relative to global estimates, and they occurred independently of gyrA/QRDR mutations, implying that their absence in panels of molecular tests for detecting fluoroquinolone resistance may yield false negative results in our setting [10]. A cross-sectional study of more than 300 patients with urinary tract infections from the Gulu Regional Referral Hospital in Uganda showed that only 51% of the isolates were sensitive to ciprofloxacin, but 67% were sensitive to levofloxacin [11].

Strategies considering substantial reductions, changes in antibiotic use, and stewardship programs are necessary to reverse trends of drug resistance [12]. Antibiotic resistance can be controlled by promoting the appropriate prescription of antibiotics, such as the restriction of the use of newer fluoroquinolones for the treatment of TB [13]. Formulary restrictions can also be used where access to certain antibiotics is limited to a specific disease state or restricted only to a specified category of clinicians [13,14,15].

The World Health Organization recognizes that data on administration can be used to relate the exposure of antimicrobials with the development of antimicrobial resistance [16]. Routine measurement and display of consumption information to prescribers and policy-makers is the first step in increasing the awareness and importance of careful antibiotic use [17].

Limited information about the consumption patterns of fluoroquinolones and the extent of their use in the management of bacterial infections other than TB is available in Uganda. The aims of this study were: (1) to document the consumption patterns of levofloxacin and moxifloxacin among selected hospitals for two years, and (2) assess the appropriateness of their use according to best practices and national treatment guidelines.

## 2. Results

### 2.1. Consumption by Outpatients, In-Patients, and TB Wards

Table 1 describes the consumption of levofloxacin and moxifloxacin during the study period, split by consumption in outpatients, in-patients, and patients treated in specialized TB wards.

Regarding outpatient consumption, two hospitals consumed levofloxacin in 2017 and three in 2018. Levofloxacin consumption decreased in 2018 in one hospital (Eastern Regional Referral Hospital, ERRH), but increased in Central Regional Referral Hospital (CRRH), the only center which, in 2018, also started the consumption of moxifloxacin. Thus, the total daily outpatient consumption of both fluoroquinolones in CRRH was 0.1 Defined Daily Doses /1000 inhabitants/day (DID).

Different figures were observed for in-hospital consumption. Moxifloxacin was not prescribed in any center, but all four hospitals prescribed levofloxacin both in 2017 and 2018, showing a significant increase in all hospitals in 2018, especially in CRRH, where consumption increased from 2.0 Defined Daily Doses /100 bed-days (DBD) in 2017 to 3.40 DBD.

Finally, both fluoroquinolones were consumed in the four hospital TB wards. As shown in Table 1, the highest consumption of levofloxacin and moxifloxacin in 2018 was observed in Western Regional Referral Hospital (WRRH) (0.3166 DID) and ERRH (0.2981 DID).

### 2.2. Appropriateness of Treatment at the Patient Level

In the second part of the study, we purposively reviewed 371 in-patient medical files of patients who received a prescription of any of the fluoroquinolones of interest across the four hospitals (WRRH = 28 files, ERRH = 83, South-West Referral Hospital [SWRRH] = 153 and CRRH = 107) from 2017 to 2018. Levofloxacin was the only third-generation fluoroquinolone prescribed in all the medical files sampled. The majority of patients were female (*n* = 194, 53.7%), and most patients were between 30 and 45 years (30.8%, *n* = 109) of age. Patients were on treatment for a median of five days (IQR = 3–5) and were prescribed a median of six doses (IQR = 4–10).

### 2.3. Adherence to Treatment Guidelines

A total of 371 prescriptions had levofloxacin; these prescriptions were filled outside TB wards. The most common indications were urinary tract infections (9.4%) followed by post-surgical prophylaxis (7.3%) and sepsis (6.7%), as seen in Table 2. Notably, all these indications are not listed for levofloxacin in the Uganda National treatment guidelines.

Table 3 describes the distribution of the appropriate and inappropriate prescriptions of levofloxacin (n = 371). Considering WHO indicators and BNF recommendations, only 220 (59.3%) prescriptions were appropriate considering the six indicators: indication, dosage, duration of treatment, formulation, appropriate age, and concomitant medicines. All the facilities involved in the study scored below the WHO recommended threshold for each indicator, except the intake and formulation of concomitant antimicrobials, which were appropriate in almost all cases (360; 97.0% and 362; 97.6%). The indication was inappropriate in 32.7% of the cases (121), followed by inappropriate dosage (14.6%) and duration of therapy (13.5%). Figure 1 describes the proportion of inappropriate prescriptions per indicator and hospital.

A Pearson Chi-square test was used to test the association between rational use of fluoroquinolones and patient sex, patient age, and hospital location. There was no significant difference in the irrational prescriptions between men and women (χ^2^ = 2.197, *P* = 0.086). By age category, 45.5% of prescriptions were irrational among the patients under 30 years of age, 50.5% among patients between 30 and 45 years old, 43.5% among patients between 46 and 65 years of age, and 29.0% among patients aged over 65 years. This difference in proportions was statistically significant (χ^2^ = 7.126, *P* = 0.046). The difference in the irrational prescriptions by hospitals was not statistically significant at the 5% level (χ^2^ = 3.705, *P* = 0.295), as ERRH and CRRH had higher inappropriate prescriptions (Table A1).

## 3. Discussion

Routine measurement of medicine consumption is a starting step in increasing awareness and antimicrobial stewardship among prescribers and policymakers. The calculation of consumption using metrics such as the number of DID or DBD can provide information about consumption and facilitate comparisons across different hospitals or treatment centers within a hospital [17]. To the best of our knowledge, no study has documented the consumption and use of third-generation fluoroquinolones in the management of bacterial infections other than tuberculosis use in Uganda.

This pilot study provides baseline data about the consumption of the newer generation fluoroquinolones (levofloxacin and moxifloxacin) using the Anatomic Therapeutic Chemical Classification (ATC) system and the DDD methodology as well as assessing the level of compliance to guidelines across four public tertiary hospitals in Uganda. After conducting two analyses, consumption according to hospital pharmacy data was determined and a detailed study of the medical charts of the patients who received fluoroquinolone was carried out.

The main result of our study identified an increase in the use of levofloxacin within the analyzed period, not only for the management of MDR/XDR-TB, but also for the management of other bacterial infections. In addition, the observed increase varied across hospitals and service centers. For example, we saw a high yearly percentage increase in levofloxacin consumption in the TB unit of the CRRH (370%), while there was another high percentage increase in the in-patient departments at WRRH and ERRH (231% and 129%, respectively). The percentage increases are consistent with a study conducted in Spain that noted a 378% growth in the consumption of levofloxacin and moxifloxacin combined between 1999 and 2002 [18]. Another study in the USA reported an increase in the use of fluoroquinolones in outpatient clinics from 1995 to 2002 [19]. The European Surveillance of Antimicrobial Consumption also noted an increasing trend in the consumption of levofloxacin from 1997 to 2009 [20]. This notwithstanding, looking at the actual in-patient hospital consumption rates, our study generally showed lower DBD in the different service centers compared to what was reported in studies conducted in similar settings. A case in point is a study at a tertiary healthcare center in India [21] that noted a consumption of levofloxacin of 62.7 DBD.

Notably, moxifloxacin was not used in the management of bacterial infections other than TB, except for one hospital, which used it in an outpatient setting. The consumption of moxifloxacin in hospital TB wards began in 2018. This could probably be explained by the erratic supply of moxifloxacin in the country due to its high price compared to levofloxacin and this could be the reason why it was not prescribed in 2017. In addition, the treatment protocols for MDR-TB provided for the use of levofloxacin until they changed in 2019 to favor moxifloxacin.

The analysis of guideline adherence showed that all prescriptions were not in line with the national treatment guidelines since medicines under study were exclusively indicated for MDR/XDR-TB in Uganda. However, in this study, all the centers were tertiary-level hospitals with chances of treating more serious conditions warranting the use of the studied fluoroquinolones; this is why such compliance was also measured based on other recommendations, such as the British National Formulary (BNF). Overall, the present analysis showed a 60% appropriate use of levofloxacin according to international recommendations (but not in accordance with Uganda guidelines), which is lower than results obtained from other studies conducted elsewhere [22]. Non-compliance varied across the study hospitals, although at a level that was not statistically significant.

Inadequate dosing (15%) and administration frequency (13.5%) of levofloxacin were among the other reasons for inappropriate use. So, considering the six indicators proposed by the WHO and the indications of use recommended by the BNF, levofloxacin was appropriately prescribed in 220 cases (59.3%). This is especially relevant as fluoroquinolones are classified in the “Watch” category of the WHO Access, Watch a Reserve (AWaRe) classification [23], a simple classification that could be used in hospital antibiotic stewardship activities to monitor or compare antibiotic use between and within hospitals. A similar study in Lebanon [22] also reported a high level of inappropriate dosing of the studied antibiotics (25%). Based on these findings, interventions to improve the use of antimicrobials should focus on different aspects of prescription and not only the indication of use to impact on its misuse and, hence, drug resistance.

The interpretation of the study results is subject to some strengths and limitations. Firstly, the use of the ATC/DDD metrics can provide a comparison in consumption information among hospitals and enables attempts to define levels of optimal use and subsequent monitoring of the medicines under study in the future. As we examined the consumption and use of fluoroquinolones in four Ugandan public hospitals, the results observed herein may not reflect consumption and use in different settings or even in private hospitals. In addition, data were collected retrospectively from manual dispensing and prescription records, and this information was used as a proxy for consumption. This method could have led either to an overestimation in cases where what was prescribed had not been actually taken by the patient, or to an underestimation if some prescriptions were dispended outside the hospital pharmacy. Nevertheless, we think that these well-known limitations of drug utilization studies did not affect the time evolution that was observed in the fluoroquinolones of interest in this study.

## 4. Materials and Methods

### 4.1. Study Setting

Uganda’s healthcare sector is composed of both public and private sectors. The public health system is organized in a hierarchical manner from the specialized national and regional referral facilities to sub-district facilities that offer primary health care. All services including essential medicines are offered free to patients. Regional referral hospitals are structured each to serve a catchment population of about 2.3–3 million people in a country with a total population of 45 million people. The services offered by the referral hospitals include outpatient care, inpatient care, and several specialty clinics including TB treatment centers. Hospitals operate private pharmacies to offer drugs not available at the general hospital pharmacy. This study was conducted in referral hospitals representing four of the countries’ administrative geographical regions. The medicines under study were provided free of charge to patients managed in the TB unit, whereas other patients prescribed fluoroquinolones would access them at the private pharmacy run by the hospital.

### 4.2. Study Design/Description

A retrospective drug utilization study was designed to analyze data on the consumption of levofloxacin and moxifloxacin in four high-volume public regional referral hospitals located in the different geographical regions of Uganda that offer specialized care including management of patients with tuberculosis. The selection was purposive based on their respective patient volumes as per the MoH annual health sector performance report [24], geographical presentation, and ease of access due to logistical challenges. This allowed for a more representative picture of fluoroquinolones use and comparison in facilities of a similar set up.

This study was conducted in four public regional referral hospitals in southwestern, western, eastern, and central regions. The study facilities included were: Western Regional Referral Hospital (WRRH) with an estimated population of 147,974 and 313 beds, Eastern Regional Referral Hospital (ERRH) with an estimated population of 297,160 and 293 beds, South West Regional Referral Hospital (SWRRH) with an estimated population of 17,428 and 314 beds, and Central Regional Referral Hospital (CRRH) with an estimated population of 58,666 and 330 beds.

The consumption was analyzed from two different sources: hospital pharmacy sales to identify all patients with and without TB receiving fluoroquinolone, and medical charts of patients without TB being prescribed fluoroquinolone. According to Uganda guidelines, third-generation fluoroquinolones are only approved for use in TB patients and prescription for TB follows strict protocols and is used for a long period of even up to 12 months. The first part of the study allowed us to identify patients without TB who received fluoroquinolone. Thus, in the second part, we studied the medical records of patients without TB using fluoroquinolone to focus on these prescriptions non-adherent to country guidelines and be able to start a stewardship program, and consider whether revising the guidelines is necessary.

### 4.3. Study Variables

The most commonly used measure for reporting medicine utilization is the number of defined daily doses (DDDs) [17]. A DDD is the assumed average maintenance dose per day for a drug used for its main indication in adults; it is a technical unit of use and does not necessarily reflect the recommended or average prescribed daily dose. However, it is a useful metric that allows for comparisons within hospitals. Thus, the consumption of the study antibiotics (levofloxacin and moxifloxacin) is expressed in terms of DDD/100 hospital bed-days (DBD) and DDD/1000 inhabitant-days (DID), considering the reference population for each respective hospital. The value of the DDD for each medicine was the reference value defined by the WHO Collaborating Centre for Drug Research and Statistics [25]: for levofloxacin oral or parenteral, 1 DDD = 0.5 g; for moxifloxacin, 1 DDD = 0.4 g.

Consumption data were obtained from the dispensing registers and prescription charts; these data sources provided proxy estimates of antibiotic use with the assumption that all drugs dispensed in a facility were consumed. Antimicrobial use data estimates were derived from patient prescription files, which allowed for disaggregation based on patient characteristics (such as sex or age), or indications for which the medicine was prescribed. Since the study medicines were rarely prescribed, all the inpatient medical charts in each facility were sampled to identify the fluoroquinolones of interest.

### 4.4. Calculation of Consumption Rates

Consumption estimates are expressed in terms of the number of DID. The number of patients treated in a year by each hospital was considered as the denominator and these population estimates were obtained from the Uganda Health Management Information System (HMIS) [24]. Thus:Consumption = (DDDs × 1000)/hospital patients in a year × 365.

Consumption rates for each service center were expressed as follows: for inpatients, as DDD per 100 bed-days; for outpatients and TB units, as DIDs.

### 4.5. Appropriateness of Use

The appropriateness of use of each antibiotic of interest consumed was assessed according to the alignment of their use with the recommended best practices and the Ugandan National Clinical Practice Guidelines [4]. Prescriptions meeting all defined criteria were classified as “appropriate” and in the absence of even one criterion, the prescription was classified as “inappropriate”

As an international reference of recommended uses of fluoroquinolone but not to assess appropriateness of use, the 2019 edition of the British National Formulary was used [26]. The WHO indicators of use and the thresholds defined by each one [27] were used as a reference for evaluation of the appropriateness of use of medicines for each prescription.

### 4.6. Data Processing and Analysis

All the collected data were entered into a common Microsoft Excel 2013 sheet by data collectors and validated before analysis. A descriptive comparison in consumption estimates between the five hospitals was made as well as individual hospital consumption for the service centers (inpatient, outpatient and TB unit). A Pearson Chi-square test was used to test the association between rational use of fluoroquinolones and the demographic characteristics of patients at 95% confidence interval.

## 5. Conclusions

This is one of the first studies to report local data about the consumption and use of the new fluoroquinolones in health facilities in Uganda. The findings revealed an increased use of fluoroquinolones, especially levofloxacin, for the management of common bacterial infections other than MDR/XDR-TB. Although we observed that, on average, less than 1% of the population served by each hospital was prescribed fluoroquinolone, a remarkable threefold increase in their use was detected during the study period. Non-compliance with the Ugandan clinical guidelines related to the prescription of levofloxacin among the clinicians was observed, a situation that may create unwarranted exposure of patients to fluoroquinolone-associated side effects as well as contribute to the appearance of antimicrobial resistance. WHO classifies fluoroquinolones as “Watch” antibiotics that should be used with caution due to their potential of developing antimicrobial resistance. Therefore, the antimicrobial stewardship programs in public facilities need to be strengthened in Uganda, as the findings showed significant differences in the irrational use of levofloxacin.

## Figures and Tables

**Figure 1 antibiotics-09-00439-f001:**
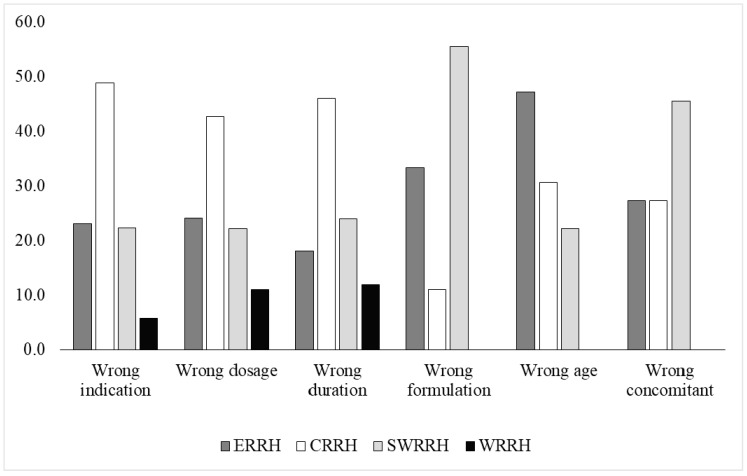
Proportion (%) of inappropriate prescriptions according to each WHO indicator (see Methods) and hospital: Eastern Regional Reference Hospital (ERRH), Central Regional Reference Hospital (CRRH), South Western Regional Reference Hospital (SWRRH), and Western Regional Reference Hospital (WRRH).

**Table 1 antibiotics-09-00439-t001:** Evolution of the consumption of levofloxacin and moxifloxacin in the different wards of the studied hospitals in 2017 and 2018.

Consumption Center (Units)	Hospital Name	Levofloxacin		Moxifloxacin
		2017	2018	2018
OPD (DDD/1000 inhabitant/day) ^1^	SWRRH	-	-	-
	ERRH	0.0024	0.0009	-
	WRRH	-	0.0013	-
	CRRH	0.0354	0.099	0.0005
IPD (DDD/100 bed-days) ^1^	SWRRH	0.643	1.471	-
	ERRH	0.78	1.37	-
	WRRH	0.026	0.086	-
	CRRH	2.00	3.40	-
TB ward (DDD/1000 inhabitant/day) ^2^	SWRRH	0.1324	0.0852	0.0852
	ERRH	0.0555	0.2611	0.0370
	WRRH	0.1676	0.2796	0.0370
	CRRH	0.1527	0.1583	0.0704

SWRRH = South West Regional Referral Hospital, ERRH = Eastern Regional Referral Hospital, WRRH = Western Regional Referral Hospital, CRRH = Central Regional Referral Hospital, OPD = Outpatient department, IPD = Inpatient department, TB = tuberculosis, DDD = Defined Daily Doses. ^1^ Computed from data collected from dispensing log book of IPD and OPD. ^2^ Computed from pharmacy stock card.

**Table 2 antibiotics-09-00439-t002:** Top 10 indications of use for levofloxacin in the studied prescriptions (*n* = 371) and proportion adhering to the indications of use included in the British National Formulary (BNF).

Indication for Use	Prescriptions, *n* (%)	Adherence to the BNF Indications of Use ^1^, *n* (%)
Urinary tract infection	35 (9.4)	35 (9.4)
Post surgical prophylaxis	27 (7.3)	6 (1.6)
Septicemia	25 (6.7)	25 (6.7)
Enteric fever	22 (5.9)	6 (1.6)
Pneumonia	14 (3.8)	14 (3.8)
Cystitis	12 (3.2)	12 (3.2)
Septicemia	12 (3.2)	12 (3.2)
Gastritis	10 (2.7)	10 (2.7)
Pyelonephritis	10 (2.7)	10 (2.7)
Pelvic inflammatory disease	10 (2.7)	10 (2.7)
Other	194 (52.3)	110 (29.6)
Total	371 (100)	250 (67.3)

^1^*N* = 371 prescriptions (100%).

**Table 3 antibiotics-09-00439-t003:** Distribution of the reasons for inappropriate use of levofloxacin in 371 studied prescriptions (100%) according to the British National Formulary and also WHO recommended indicators and thresholds (see Methods).

Indicator (WHO Recommended Threshold)	Inappropriate, *n* (%)	Appropriate, *n* (%)
Indication of use (90%)	121 (32.7)	250 (67.3)
Dosage (95%)	54 (14.6)	317 (85.4)
Duration (95%)	50 (13.5)	321 (86.5)
Formulation (95%)	9 (2.4)	362 (97.6)
Age (95%)	36 (9.7)	335 (90.3)
Concomitant medicine intake (95%)	11 (3.0)	360 (97.0)

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
