# Peer review of "Prescription of Levofloxacin and Moxifloxacin in Select Hospitals in Uganda: A Pilot Study to Assess Guideline Concordance"

_antibiotics, 2020, doi:10.3390/antibiotics9080439_

Round 1

Reviewer 1 Report

This is a retrospective analysis of fluoroquinolone use in four Ugandan hospitals/areas and an assessment of if this use adhered to national and international guidelines. Overall, the manuscript gives an overview of how these antimicrobials are being misused in one country, which may limit the interest in readership outside of that one country and/or surrounding countries. There are some issues with the manuscript as detailed below.

Throughout the manuscript, some words are lacking a space between them. See lines 214-215 for a few (of the many) incidents of this. There are also some mis-spellings and grammar issues (see lines 250-251 for examples).

Title: Right now, it gives away the results of the study. Recommend removing everything after “pilot study” or change to something like “Prescription of levofloxacin and moxifloxacin in select hospitals in Uganda: A pilot study to assess guideline concordance”

Introduction: There is a lot of general information in this section that are tangential to your topic. For examples, lines 51-57 could be removed without losing a significant piece of background information.

Results, lines 69-73: The first paragraph would fit better in the Discussion section. Recommend moving or deleting.

Results, lines 78-81: To better orient the reader, you might want to include how outpatient prescriptions are assigned to hospitals in the Methods section. In other countries, most outpatient prescriptions would not come from a “hospital” but rather a clinic or physician office. It also is not clear if the TB ward is a part of each hospital, so recommend clarifying that as well. Another helpful clarification would be if patients with TB could be treated outside of the TB ward or if they had to be in the TB ward to be treated.

Results, Table 1: Why wasn’t moxifloxacin used at all in 2017 in any facility? (might be best addressed in the Discussion section)

Results, Table 1:  Having the change in levofloxacin use as a column is confusing. Consider reformatting in a different way (maybe as a row beneath).

Results, Line 110: Why were no levofloxacin prescriptions from the TB ward evaluated? This seems like a major flaw in the study, especially if it’s the case that patients are not (or are rarely) treated for TB outside of a TB ward.

Materials and Methods: How were these hospitals selected? You say it’s due to patient volumes (did you want larger facilities or smaller ones and why?) and distance between each other (did you want facilities far from each other or close and why?). Please clarify.

Materials and Methods: Why were 3 different sources used to determine appropriateness? Please explain.

Author Response

Please see the attachment with responses. 

Reviewer 2 Report

Introduction – I suggest adding more information about antimicrobial resistance to levofloxacin in Uganda. Are there any studies regarding this subject, which might be correlated with the antibiotic prescription and consumption data from the results and compared in the discussion part?

It is not clear when did the national guidelines for tuberculosis treatment changed, was it in 2017, since the authors decided to evaluate the fluoroquinolone use in 2017-2018?

Line 56-57 – the cited study identified an increased resistance in Uganda? How did they reached this conclusion, after analysing a small group, a hospital, or epidemiological available data?

Results – The results are very limited. I suggest adding more data, if available, or trying to analyse them more, in order to get a better picture. Perhaps adding some figures.

Table 2 – I would change “septicaemia” with ”sepsis”

Materials and methods Line 208-217 – it is not very clear in what proportion do the 4 healthcare settings serve the population of Uganda. In order to better understand, the reader must look the population of the country and calculate how much of it is it attended in those settings in order to get a picture of the amplitude of the results.

It is not very clear, in Uganda antibiotics can not be acquired from other pharmacies except the ones from the hospital? The evaluation of the antibiotic prescription and acquiring might be underestimated since the hospitals might not be the only source.

In Uganda, the legislation permits buying antibiotics only based on a prescription? If so, can’t anyone buy even if it doesn’t have a prescription?

Line 220 – I suggest to be reformulated; in the present form it suggests that the study was conducted in 7 regions.

As a general aspect, please revise the spelling and expression – there are numerous words not well written for example – line 251 “inapropraite”, line 275 ”a significant differences”, etcetera

Author Response

Kindly see attachment. 

Round 2

Reviewer 1 Report

This revision is much better than the original submission. I appreciate all of the changes the authors made. I think it is acceptable to publish.

Reviewer 2 Report

I do not have other comments.